# Functional Genomics of a Collection of Gammaproteobacteria Isolated from Antarctica

**DOI:** 10.3390/md22060238

**Published:** 2024-05-23

**Authors:** Michele Giovannini, Walter Vieri, Emanuele Bosi, Christopher Riccardi, Angelina Lo Giudice, Renato Fani, Marco Fondi, Elena Perrin

**Affiliations:** 1Department of Biology, University of Florence, Via Madonna del Piano 6, I-50019 Sesto Fiorentino, Italy; michele.giovannini@unifi.it (M.G.); walter.vieri@unifi.it (W.V.); christopher.riccardi@unifi.it (C.R.); renato.fani@unifi.it (R.F.); marco.fondi@unifi.it (M.F.); 2Department of Earth, Environment and Life Sciences—DISTAV, University of Genoa, Corso Europa 26, I-16132 Genova, Italy; emanuele.bosi@unige.it; 3Quantitative and Computational Biology Department, University of Southern California, Los Angeles, CA 90089, USA; 4Institute of Polar Sciences, National Research Council, (CNR.ISP), Spianata San Raineri 86, I-98122 Messina, Italy; angelina.logiudice@cnr.it; 5Italian Collection of Antarctic Bacteria, National Antarctic Museum (CIBAN-MNA), I-98122 Messina, Italy; 6NBFC, National Biodiversity Future Center, Piazza Marina 61, I-90133 Palermo, Italy

**Keywords:** Antarctic bacteria, Gammaproteobacteria, genome sequencing, secondary metabolite biosynthesis, phylogenomic analysis, cold-shock proteins, environmental preservation

## Abstract

Antarctica, one of the most extreme environments on Earth, hosts diverse microbial communities. These microbes have evolved and adapted to survive in these hostile conditions, but knowledge on the molecular mechanisms underlying this process remains limited. The Italian Collection of Antarctic Bacteria (*Collezione Italiana Batteri Antartici* (CIBAN)), managed by the University of Messina, represents a valuable repository of cold-adapted bacterial strains isolated from various Antarctic environments. In this study, we sequenced and analyzed the genomes of 58 marine Gammaproteobacteria strains from the CIBAN collection, which were isolated during Italian expeditions from 1990 to 2005. By employing genome-scale metrics, we taxonomically characterized these strains and assigned them to four distinct genera: *Pseudomonas*, *Pseudoalteromonas*, *Shewanella,* and *Psychrobacter*. Genome annotation revealed a previously untapped functional potential, including secondary metabolite biosynthetic gene clusters and antibiotic resistance genes. Phylogenomic analyses provided evolutionary insights, while assessment of cold-shock protein presence shed light on adaptation mechanisms. Our study emphasizes the significance of CIBAN as a resource for understanding Antarctic microbial life and its biotechnological potential. The genomic data unveil new horizons for insight into bacterial existence in Antarctica.

## 1. Introduction

Microorganisms were the first inhabitants of the Earth, and since then, they have spread throughout the planet, adapting to live in a wide range of ecological niches (including environments where they are the only form of life) [1,2]. It has been estimated that today our planet is populated by about a trillion species of microorganisms [3], most of which are useful and essential to life [2]. Contingent on the colonization of a remarkable variety of ecological niches, including harsh environments, they have been evolving genes that allow them to perform many functions on which we currently depend. Indeed, we rely on microorganisms to make food and to digest it in our gut, produce antibiotics, treat waste or to fix atmospheric nitrogen, just to give a few examples [2].

They represent an invaluable resource for research and for applications in different fields spanning from industrial to agricultural to health care. For example, they can carry out at room temperature chemical reactions that we could only perform at extremely high temperatures with serious environmental costs. In addition, they can be engineered to produce biological molecules, drugs, and enzymes and they also provide genetic capacities that drive modern gene technologies (such as plasmids and CRISPR/Cas systems used for eukaryotic genome editing) [1,2].

For all these reasons and more, monitoring and preserving microbial diversity is of fundamental importance [1]. In this context, bacterial culture collections are indispensable, allowing for the conservation of the original bacterial isolates, but also the accessibility of microbial strains and the related genetic materials to the scientific community for research and applications [1]. Since 1998, the Organisation for Economic Co-operation and Development (OECD) recognized the role of culture collections in the future of life sciences [4].

In addition, the public availability of bacterial gene and genome sequences is crucial, both for basic and applied science. Genomic data serves as a foundational element within systems biology methodologies. These methodologies integrate genomics, transcriptomics, proteomics, metabolomics, bioinformatics, mathematics, and other relevant disciplines to comprehensively elucidate the intricate mechanisms underlying cellular function. Besides offering a more integrated perspective on the working scheme of a cell, systems biology also has a wide range of applications. For example, in bacteria, systems biology approaches have been used for metabolic engineering of bacterial strains [5], for synthetic biology [6], and for drug discovery pipelines [7]. An important point for a systems biology approach is the availability of -omics data on the microbial strains of interest, the possibility to store and retrieve the large amounts of data that are produced during this type of study, and also to facilitate the access to all the information for other researchers. In recent years, the importance of the single microbial genes has been highlighted since they encode the majority of the functional repertoire of life on earth. In 2022, the creation of the non-redundant Global Microbial Gene Catalogue (GMGCv1) revealed that most bacterial genes are rare and habitat specific [8], while in 2024 the creation of the KAUST Metagenome Analysis Platform (KMAP) Global Ocean Gene Catalog 1.0 represents the largest open-source framework to date, matching microbial classes with gene function, geographic location, and ecosystem type [9].

Antarctica is one of the most hostile environments in the word. It is almost totally covered by glacier ice and can be divided into two main areas, the East and West Antarctica, which are physically separated by Transantarctic Mountains and characterized by a different thickness of the ice layer. Much of the continent’s coastline is fringed by ice shelves, which, depending on the seasons, can cover the several large and small islands surrounding the continental part (https://discoveringantarctica.org.uk/, accessed on 7 May 2024). 

Antarctica is distinctive for being the coldest (with temperatures that can reach −30 °C), windiest (winds up to 327 km h^−1^), and the driest (only 200 mm of precipitation per year) continent (https://discoveringantarctica.org.uk/, accessed on 7 May 2024). Its climate is characterized by large variations across the continent owing mainly to differences in latitude, altitude, and distance from the Southern Ocean. In particular, some coastal areas have micro-climate and topographic conditions, which during the austral summer, cause enough melting to allow some land to remain free of glaciers (https://discoveringantarctica.org.uk/, accessed on 7 May 2024) [10].

Also, the Southern Ocean is subject to seasonal fluctuations in temperature (from −2 to 10 °C), which is related to the seasonal advance and retreat of sea ice [11,12,13]. Nonetheless, both Antarctic aquatic environments (sea, sea-ice, and lakes from freshwater to highly saline) and soils are inhabited by highly diverse microbial communities [10]. These circumstances make Antarctica a reservoir of undiscovered microorganisms and novel genes and molecules that can be used not only to understand the adaptation of cells to extreme environments, but also to develop new biotechnological products [10].

Since 1989 the Department of Chemical, Biological, Pharmaceutical and Environmental Sciences of the University of Messina manages the Italian Collection of Antartic Bacteria (Collezione Italiana Batteri Antartici (CIBAN), included in the Museo Nazionale dell’Antartide (MNA)) that embeds cold-adapted strains collected in different areas of Antarctica during seven Italian Expeditions and represents one of the few collections in the world dedicated to Antarctic bacteria. Sampling mainly regarded marine and lacustrine environments through the collection of different environmental matrices, both abiotic (water (both at surface and along the water column) and sediment) and biotic (specimens of Porifera). Most isolates have been obtained from two Antarctic sites: the Terra Nova Bay (Ross Sea) and the Antarctic Peninsula. Bacterial strains of the CIBAN-MNA have been identified by sequencing their 16S rRNA genes and were phenotypically characterized. Members of the class Gammaproteobacteria are predominant, followed by Actinobacteria, Alphaproteobacteria, Bacteroidetes, and to a lesser extent, by Firmicutes and Betaproteobacteria.

This collection represents an important resource for the study of (i) bacterial life in Antarctica, (ii) cold adaptation in general, (iii) evolutionary trajectories, and (iv) potential biotechnological applications. Several research works have been published during the years on the strains belonging to this collection [14,15,16,17,18,19,20,21,22,23,24,25,26,27,28,29,30], and the CIBAN-MNA collection represents an important resource for the study of Antarctic bacteria. Currently, however, no -omics data for this collection are available despite the potential for such information, combined with a systems biology approach, to allow a better understanding of the biology of these strains, of their metabolisms, of their adaptation to the extreme Antarctic condition, and also their future abilities to face climate change. Moreover, these data could be used for a lot of different applications including the identification of new bioactive molecules produced by these strains (e.g., antibiotics, anticancer, or biosurfactants [31]). Finally, a more comprehensive view of the mechanisms of resistance to different types of stress (e.g., heavy metals or antibiotics) could be obtained. 

With the aim to valorise the CIBAN-MNA collection of bacteria isolated from Antarctica, we sequenced, analyzed, and made publicly available the genome sequences of 58 marine Gammaproteobacteria strains belonging to the aforementioned collection. All the strains were isolated from Terra Nova Bay in the course of several Italian expeditions to Antarctica (between 1990 and 2005).

## 2. Results and Discussion

### 2.1. General Genome Features and Creation of Custom Databases

The average nucleotide identity (ANI) map of 64 CIBAN-MNA samples revealed four distinct clusters, consistent with the genera *Pseudomonas*, *Pseudoalteromonas*, *Shewanella*, and *Psychrobacter*. The robustness of these clusters was corroborated by the inclusion of NCBI genomic reference sequences nested within each respective bacterial genus (refer to Figure 1). A significant part of our analysis aimed at elucidating the taxonomic classification of the newly assembled genomes, previously classified using 16S rRNA sequences (for references see Appendix A). Notably, ANI analysis revealed some inaccuracies in previous taxonomic assignments solely based on 16S sequences. Genomes with incorrect 16S taxonomic assignments were reclassified with the appropriate bacterial genus names (Appendix A). Furthermore, genomes outside the four main clusters underwent taxonomic reassignment. *Marinomonas* strains W1-45 and E12 remained in agreement with the assignments based solely on 16S analysis, forming a distinct cluster (Figure 1). In the taxonomic reassessment, *Colwellia* GW185 was reclassified as belonging to the genus *Bacillus* and *Alteromonas* strain GW104 as part of the genus *Psychromonas* (Figure 1). In addition, based on the number of contigs, three genomes were excluded due to excessive fragmentation (Appendix A). Moreover, to avoid numerical imbalance among the different genera and to exclude taxonomic outliers from the dataset, genomes that fell outside the main clusters were omitted from downstream analyses. Consequently, from the comprehensive set of 64 sequenced and annotated genomes, 58 genomes, emblematic of central clusters, were systematically preserved for further analysis. The general features of these genomes are reported in Table 1. In order to increase the robustness and the representativeness of each genus in the downstream analyses, we incorporated 234 representative genome sequences from the NCBI and included it in our final genomic dataset. The number of genomes for each database created is shown in Appendix A.

### 2.2. Phylogenomic Analyses and Functional Content of the Genomes

We investigated interspecies phylogenetic relationships among the newly sequenced bacterial genomes by exploring the phylogenomic trees in Figure 2 inferred from a set of core genes (see Appendix A for bootstrap supports of all nodes and the Section 3 for details on the preparation and inference of all trees). The phylogenomic relatedness within the genus *Psychrobacter* was investigated utilizing a concatenation of 107 core genes. The resulting tree divided *Psychrobacter* into two main clades (Figure 2A): one consisting of organisms isolated from warm hosts (such as human blood, seal feces, and a tropical marine fish), and the other comprising organisms predominantly found in terrestrial or marine low-temperature environments. The topology appeared robust as most nodes were supported by 90%–100% of the tree (Appendix A). In detail, the clade of organisms associated with warm hosts includes *P. pasteurii*, *P. phenylpyruvicus*, *P. piechaudii*, *P. sanguinis*, *P. lutiphocae*, *P. pygoscelis*, and *P. arenosus*. Whereas the remaining organisms in the analyzed dataset, including the 15 newly sequenced genomes, were grouped within the other main clade. Within this clade, the placement of the new genomes appeared scattered, except for the group consisting of *Psychrobacter* strains HY3, CAL606, GW208, CAL495, GW64, and 78a. For the *Pseudomonas* genus, our phylogenetic analysis was based on a concatenation of 43 core genes. In the resulting tree (Figure 2B), *Pseudomonas* organisms formed two primary clades, each further subdivided into distinct clades/subgroups. The majority of external nodes exhibit robust support as indicated by high bootstrap values (Appendix A). However, not all nodes display such clarity, as the *Pseudomonas* genus is characterized by extensive genetic diversity and genomic plasticity [32,33]. Therefore, the tree reveals particularly high genetic heterogeneity within the *Pseudomonas* genus. All our new genomes were clustered within the same clade, except for *Pseudomonas* strain E45, which was placed in another distinct main clade, indicating a greater phylogenetic distance. The concatenation of 92 core genes specific to the *Pseudoalteromonas* genus resulted in a tree with consistently high support values across all nodes, including both external and internal branches (Figure 2C). The newly obtained genomic sequences of *Pseudoalteromonas* formed three distinct subgroups, closely clustered together phylogenetically, belonging to the same major monophyletic clade. Phylogenomic analysis of the 74 concatenated core genes of the *Shewanella* genus revealed that the new organisms were placed in two closely related groups, supported by robust bootstrap values (Figure 2D and Appendix A). These groups belong to the same clade, which encompasses other bacteria found in polar environments, including *S. polaris*, *S. psychromarinicola*, and *S. frigidimarina* [34], suggesting their potential common adaptation to Antarctic environmental conditions.

After a thorough taxonomic classification of the strains, we investigated their genome-level functional content. The clusters of orthologous group (COG) annotation was conducted using eggNOG-mapper, resulting in a matrix detailing the relative abundance of broad functional categories within each sequenced and assembled genome. As expected, the variance of functional categories exhibited a consistent pattern across each genus (Figure 3), revealing a uniform distribution of metabolic functions across the genome dataset. Excluding the S category (function unknown) some categories showed higher variance than the others. Notably, categories C (mean = 6.41; standard deviation SD = 1.09), involved in energy production and conversion, and K (mean = 6.38; SD = 1.22), responsible for transcription, displayed the highest variability. Additionally, categories L (mean = 5.79; SD = 1.05), associated with replication, recombination, and repair, and T (mean = 5.93; SD = 1.64), involved in signal transduction mechanisms, showed considerable variability across the genomes. Overall, these analyses revealed a consistent dataset of Antarctic genomes, both from a taxonomical and functional viewpoint and suggested that, to better highlight their functional diversity (if any), we had to dig deeper into more specific functional categories. For this reason, we explored the pool of secondary metabolites, antibiotic resistance, and cold-adaptation genes owned by these groups of microorganisms.

### 2.3. Genome Mining of Secondary Metabolites

AntiSMASH is an invaluable resource for the automated identification and analysis of biosynthetic gene clusters (BGCs) responsible for the production of secondary metabolites in microbial genomes. As the presence of BGCs could be indicative of previously undiscovered biosynthetic potential, we mapped their presence across the CIBAN samples (Figure 2). We also studied the presence/absence patterns of the detected BGCs and applied an unsupervised machine learning clustering method to group together genomes with similar trends (Figure 4). Remarkably, the strictly numerical analysis with DBSCAN and the more sensitive grouping with BIG-SCAPE, which utilizes protein domain content, order, copy number, and sequence identity, produce overlapping results. In detail, the DBSCAN algorithm classified 53% of the data as a low-density region (cluster A), consisting of outliers or data points that did not have enough neighbor points to be part of any cluster. These genomes exhibited a distinct or “proprietary” secondary metabolite profile distribution and were assimilated into the largest cluster (Figure 4). Thus, these genomes do not share any BGC but, rather, each of them is characterized by a unique BGC profile.

Most *Pseudomonas* genomes, including both the newly sequenced genomes and those from the NCBI reference database, fell into this category (cluster A) due to their variegated patterns (Figure 4). However, two exceptions were observed. Notably, a distinct cluster was formed (cluster G), comprising two newly sequenced genomes, *Pseudomonas* strains 65/3 and HY13, along with *Pseudomonas lundensis* genome reference of NCBI (Figure 4). These three genomes exhibited a common pattern of BGCs, including arylpolyene, redox cofactor, N-acetylglutaminylglutamine amide (NAGGN), non-ribosomal peptide synthetase (NRPS), betalactone, and RiPP-like metabolites (Figure 2, Appendix A). Redox cofactor, NAGGN, and betalactone showed a strong prevalence in *Pseudomonas* compared to other genera (Appendix A). The final exception was observed in cluster H (Figure 4), where *Pseudomonas pharmacofabricae* showed a notably closer similarity to *Psychrobacter lutiphocae* (Figure 2). Analyzing the diversity within novel genomes of the *Pseudomonas* genus, it is noteworthy that only *Pseudomonas* strain E45 had both the terpene and ectoine clusters (Figure 2, Appendix A). Most known terpenes are derived from terrestrial sources, notably plants and fungi [35,36]. Indeed, marine terpenes have received scarce attention, with limited understanding of the biochemical processes governing their synthesis, except for a few studies on algae and marine invertebrates [37,38]. The increasing identification of terpenes from marine bacteria suggests a significant number of these compounds remain to be isolated and characterized [39,40]. The precise role of these compounds in bacterial cell physiology remains poorly understood. However, they may be involved in stress mitigation, preservation of cell membrane integrity, photoprotection, attraction or repulsion of organisms, promotion of host growth, and defense mechanisms [41]. In contrast, gene clusters coding for the biosynthesis of the ectoine are common among the marine or halophilic bacteria [42,43] as this compound helps to cope with high salt concentrations. Nevertheless, among the sequenced genomes, only *Pseudomonas* strain E45 possessed this BGC. Thus, other strains might potentially employ different organic solutes, adapting to dilution stress or sudden increases in salinity [44]. Among the newly sequenced genomes, another notable observation is that only *Pseudomonas* strains E45 and HY14 contain siderophore BGCs (Figure 2, Appendix A). In the oceans, iron is often scarce, creating a limiting environment [45]. Hence, these marine bacteria can produce siderophores to satisfy their needs of iron [46]. In contrast to the *Pseudomonas* genus, several of the newly sequenced organisms within the *Pseudoalteromonas* genus formed distinct groups (Figure 4). Indeed, many genomes were grouped outside of cluster A, which encompasses genomes showing greater diversity among each other. Specifically, five similarity clusters were identified, designated as clusters B, C, D, E, and F (Figure 4). In analyzing each new genome, a low number of BGCs were identified (Figure 2). Notably, the prevalent BGC types included RiPP-like clusters, arylpolyene clusters, with siderophore clusters appearing sporadically across some genomes (such as *Pseudoaltermonas* strains TB43, D48, G24, and MR144) (Appendix A). 

*Shewanella* bacteria showed a substantial number of BGCs related to arylpolyene, PUFA, hglE-KS, and betalactone (Figure 2). Numerous genomes were clustered in the largest group (cluster A) (Figure 4), highlighting the significant diversity within this genus. Indeed, the adaptability of *Shewanella* physiology enables their remarkable distribution across a broad spectrum of ecological niches [47]. Additionally, a total of eight distinct similarity clusters were identified, labeled as clusters O, P, Q, R, S, T, U, and V (Figure 4). In these clusters, genomes exhibiting similar patterns were grouped collectively. Together, these clusters highlight the remarkable diversity within this genus. Specifically, within cluster R (Figure 4), the genome of the new *Shewanella* strain T24 was grouped alongside *Shewanella psychromarinicola* [34]. Phylogenomic analyses further affirmed the close phylogenetic relationship between these two bacteria (Appendix A), both originating from sediments of Antarctic environments. For the remaining new genomes classified within clusters A and O (Figure 4, Appendix A), the most prevalent BGC class was RiPP-like, a class encompasses a diverse group of biologically active bacterial molecules [48]. RiPP-like compounds offer a promising alternative to antibiotics synthesized via polyketide or non-ribosomal pathways [49,50,51,52]. Their limited spectrum of antimicrobial activity positions them as potential candidates for clinical applications. Unlike wide-spectrum antibiotics, RiPP-like compounds reportedly mitigate off-target effects, minimizing disruptions to normal flora and reducing the risk of secondary infections by resistant organisms [52].

*Psychrobacter* exhibited the lowest number of BGCs (Figure 2) (Appendix A). The newly sequenced genomes, with the sole exception of *Psychrobacter* strain 16, possess the betalactone BGC (class = “Others”). Additionally, this specific BGC is prevalent in the majority of other *Psychrobacter* genomes obtained from NCBI (Figure 2, Appendix A). The betalactone BGC is known for its involvement in the production of antimicrobial compounds [53,54]. Siderophore BGCs were identified in several of the newly sequenced genomes, including strains W2-37, TB55, 78a, W1-15, and GW64. These molecules, along with ectoines and terpenes, play crucial biological roles in microbial community adaptations to harsh environmental conditions [55,56]. Additionally, these small molecules, aside from their role in iron acquisition, also exhibit antimicrobial activity [57]. In contrast to the other sequenced genomes, all the newly *Shewanella* genomes exhibit the presence of the polyunsaturated fatty acid (PUFA) cluster (Appendix A). PUFA products play a dual role in marine bacteria, contributing to both cold adaptation and antioxidative functions. Indeed, the presence of extreme environments with low temperatures, such as the Antarctic oceans, promotes the adaptation of bacteria that leads to the production of PUFAs. These bacteria can modify the content of hopanoids, proteins, carotenoids, sterols, and fatty acids in the cell membrane [58,59,60]. Therefore, the survival of these microbes in extremely cold habitats is facilitated by the incorporation of specific fatty acids into the membrane, enabling nutrient transport and maintaining membrane fluidity [61]. Additionally, studies have indicated that PUFAs may confer antioxidant properties in bacterial cells by shielding the membrane [62,63,64]. Thus, the cell membrane-shielding effect of PUFAs hinders the passage of exogenous hydrophilic compounds, such as H_2_O_2_, through the membrane [64].

### 2.4. Characterization of Antibiotic Resistance Genes

We used the tool RGI from the CARD to identify and classify putative antibiotic resistance genes (ARGs) present in these newly sequenced genomes. Different algorithms were used to detect antimicrobial resistance (AMR) proteins: “Perfect” for exact matches, “Strict” for slight variations within cut-off scores, and “Loose” for broader detection, including distant homologs [65]. No “Perfect” results were obtained but only few “Strict” hits for each genome. Consequently, the number of hits was increased by including “Loose” results. However, to ensure robustness, only hits showing at least 80% identity with the reference protein were considered significant and retained. Inactivation was the most abundant mechanism of action among predicted ARGs, followed by efflux mechanisms and target alteration (Figure 5). In detail, all newly sequenced Antarctic bacterial genomes exhibited resistance to multiple classes of antibiotic drugs, including fluoroquinolones, diaminopyrimidines, and phenicols. Bacteria belonging to the *Pseudomonas* genus displayed a broader spectrum of antibiotic resistance (Figure 5). This spectrum also encompassed resistance to tetracycline, a trait shared with some genomes of the *Pseudoalteromonas* genus. Indeed, *Pseudoalteromonas* and *Pseudomonas* bacteria appeared to possess a set of genes conferring resistance to a wider range of antibiotics; this trend is reversed in bacteria of the *Shewanella* and *Psychrobacter* genera (Figure 5). Moreover, nearly all bacterial strains exhibited susceptibility pattern to antibiotics belonging to the sulfonamides, aminocoumarins, and glycylcycline classes, except for those within the *Pseudomonas* genus (Figure 5). Distinct strains were also identified, showing resistance to a class of antibiotics not shared with (or shared only with few) other bacteria analyzed. Notably, *Psychrobacter* strain 16 was the only bacterium resistant to nucleoside antibiotics, while *Psychrobacter* strains EVC214 and TB20 were resistant to streptogramin B antibiotics. Although these results are derived from computational predictions of ARGs and so they would require in vitro phenotypic validation, they offer a comprehensive insight into the resistome of the newly sequenced Antarctic genomes.

While the argument has previously been made [66,67], these results emphasize the scarcity of completely “pristine” environments on Earth. Additionally, the identification of ARGs in environments with minimal human impact, such as Antarctica, could offer insight into baseline contamination levels, the extent of contamination, and how these contaminants spread within the environment. At the same time, the presence of ARGs could also suggest intricate microbial interactions in Antarctica. Indeed, antibiotics and ARGs act as both weapons and shields in bacterial warfare [68,69]. Additionally, ARGs have been identified as significant biotic factors influencing microbial interactions [70].

### 2.5. Cold-Adaptation Proteins (CAPs)

The presence of cold-adaptation proteins (CAPs) was assessed utilizing the database from a previous study conducted by Bosi et al., 2017 [71]. The Antarctic marine ecosystem exhibits low temperatures, thereby requiring resident microorganisms to possess genes associated with cold adaptation. The presence/absence analysis utilizing the CAPs protein database revealed, as expected, the presence of at least one cold-adaptation gene in all newly sequenced genomes (Figure 6). In detail, we observe the formation of two distinct gene clusters (Figure 6). The first cluster is shared among nearly half of the newly sequenced genomes and is relatively rare among *Pseudomonas* bacteria, except for the HY13 strain. The genes within this cluster encode type III antifreeze proteins (AFPs) and membrane fusion proteins (MFPs). AFPs constitute a varied category of ice-binding proteins that prevent ice formation by lowering the freezing point of a solution below its melting point [72]. This interaction leads to ice growth occurring on a curved surface between adjacent AFPs, thereby reducing the freezing point [73,74]. Cold temperatures can impact the fluidity and composition of cell membranes [75,76]. MFPs likely play a role in regulating membrane lipid homeostasis, thereby preserving the fluidity and structural integrity of cell membranes in bacteria [74,77,78].

The second gene cluster was present in nearly all genomes, with the exception of several *Pseudoalteromonas* strains (Figure 6). These genes encode AFPs type I, suggesting that their absence in some *Pseudoalteromonas* strains might be compensated by the presence of AFPs type III in the first gene cluster. Notably, *Shewanella* strains S1-49, CAL98, and T24 were the only organisms found to possess genes responsible for encoding ice-binding proteins (Figure 6). Among them, *Shewanella* strain S1-49 emerges as particularly well-equipped with genes associated with cold adaptation (Figure 6). In contrast, a subset of genomes, such as those of *Pseudoalteromonas* strains 69, CAL260, 120, 45, 43, 20, 24, 3, and 19, exhibited a reduced number of genes. Notably, these genomes only possessed the gene encoding the antifreeze protein (D0RKK3), while the remaining bacteria were adequately equipped (Figure 6). In conclusion, our analysis revealed that all newly sequenced genomes harbor at least one cold-adaptation gene, highlighting the variability in genetic adaptations within the microbial community.

## 3. Materials and Methods

### 3.1. Bacterial Strains, Media, and Growth Conditions

All the Gammaproteobacteria strains used in this work belong to the “Collezione Italiana Batteri Antartici (CIBAN)” (included in the “Museo Nazionale dell’Antartide (MNA)”). They were isolated from Terra Nova Bay during several Italian expeditions to Antarctica (between 1990 and 2005) financially supported by “Programma Nazionale di Ricerche in Antartide (PNRA)” and are conserved at the University of Messina. All the isolates are of marine origin. The list of bacterial strains and all the information related to their sampling are reported in Appendix A. All strains were routinely grown in Marine Agar (MA) or Broth (MB) (Condalab, Spain) under aerobic condition at 21 °C. The stock suspensions of the strains were stored in 20% [*v*/*v*] glycerol solution at −80 °C.

### 3.2. Genome Extraction and Sequencing

Total DNA was extracted using a DNeasy UltraClean Microbial Kit (QIAGEN S.r.l., Venlo, The Netherlands) following the manufacturer’s instructions. DNA concentration and quality were assessed using a QUBIT dsDNA Quantitation, High Sensitivity kit and a Qubit 4 Fluorometer (both from Invitrogen—Thermo Fisher Scientific Inc., Waltham, MA, USA) and an Infinite 200 PRO Tecan plate reader (Tecan Group Ltd., Männedorf, Switzerland). DNA sequencing (2 × 150 bp) was performed by BMR genomics S.r.l., Padova, Italy, on an Illumina platform (Illumina, Inc., San Diego, US). Quality control of the reads was assessed with FastQC v0.12.1 [79]. SPAdes v3.15.5 [80] was employed for the de novo assembly of the Illumina reads using the default parameters (kmers of lengths 21, 33, and 55) and the *--careful* option. A visualization of the new assemblies contiguity and completeness was generated using QUAST v5.2.0 [81]. Sequences obtained in this work are publicly accessible under the NCBI BioProject PRJNA1100444. Sequences underwent filtering to exclude contigs < 199 nt before submission to the NCBI portal.

### 3.3. Genus-Based Taxonomic Clustering and Database Construction

All assembled bacterial strains were previously taxonomically classified only using partial 16S rRNA gene sequences (for references see Appendix A). Average nucleotide identity (ANI) between genomes was calculated via FastANI v1.33 [82] (kmer size of 16 bp and fragment length of 3000 bp) to confirm the genomes relatedness of strains. In order to confirm that organisms within each identified cluster were not duplicates of the same strain, we utilized the JSpecies Web Server (JSpeciesWS) [83]. This service enabled the calculation of average nucleotide identity [84] using both BLAST+ (ANIb) and MUMmer (ANIm), as well as correlation indexes based on tetranucleotide signatures (Tetra). Genomes identified outside the main clusters via FastANI underwent taxonomic reassignment. This process involved a double-checking procedure, first based on clustering according to ANI scores, followed by confirmation using the Type (Strain) Genome Server (TYGS) [85]. Then, non-redundant databases for the different bacterial genera identified were generated. Specifically, from the NCBI portal, genomes for each genus were selectively downloaded. Due to the extensive data available on NCBI, particularly for the *Pseudomonas* genus, sequences were filtered based on the *reference_genome* and *representative_genome* categories, ensuring higher quality and representativeness. Then, to ensure balanced representation and comparability across databases while preserving diversity, the Mash distance was calculated using Mash tool v2.3 [86]. Subsequently, thresholds were established to limit the inclusion of similar genomes. For *Pseudoalteromonas*, *Psychrobacter*, and *Shewanella* genera, thresholds ranging from 0.05 to 0.3 were applied, whereas for *Pseudomonas*, characterized by a larger genome pool, thresholds ranged from 0.12 to 0.3. The subsequent analyses were conducted on all genomes within each database, encompassing both newly sequenced and assembled genomes as well as those selected and filtered by NCBI.

### 3.4. Genome Annotation and Genome Content Comparison

Putative protein-coding sequences (CDSs) and non-coding RNA genes were predicted using Prokka v1.14.6 [87] with the *--rfam* option on all genomes. Then, amino acidic sequences predicted by Prokka were used as input to a custom *bash* script using the eggNOG-mapper v2.1.12 [88] to infer functional features based on orthology prediction. The tabular output files generated were imported into R 4.2.2 to visualize the distribution of relative abundances across different functional categories within genomes. In addition, clusters of orthologous groups (COGs) identified in the genomes were compared with those from reference genomes obtained from NCBI for a comprehensive assessment of functional similarities.

### 3.5. Detection of Secondary Metabolites

Biosynthetic gene clusters (BGCs) associated with the production of secondary metabolites on all genomes were identified using AntiSMASH v6.1.1 [89] with default parameters, enabling all prediction features. The AntiSMASH results were cross-validated through BiG-SCAPE v1.1.5 [90] (using *--pfam_dir*, *--mibig, and --min_bgc_size*
5000 as parameters). Specifically, BiG-SCAPE was utilized locally to analyze the BGCs as individual .gbk files detected from the AntiSMASH tool. Subsequently, the validated outputs were processed using a Python script to generate a matrix displaying the frequency of each identified BGC. The abundance matrices of secondary metabolites detected through AntiSMASH (n = 42) were clustered using DBSCAN in the scikit-learn 1.4.1 [91] Python library (parameters: eps = 1.8, min_samples = 2). A total of 47% of the data were assigned to unique clusters of sizes 2 through 19.

### 3.6. Phylogenomic Analyses

Separate phylogenomic analyses were conducted for each genus-specific database. In detail, core genes classified using Roary v3.12.0 [92] with 90% identity for blastp, were used as molecular markers for constructing the phylogenomic tree. Subsequently, the amino acid sequences of the core genes were aligned using Muscle v3.8.1551 [93] and polished with Gblocks 0.91b [94] with default settings. Finally, all core genes were concatenated together into a single sequence. The optimal amino acid substitution model for the alignment was determined with ProtTest v3.4.2 [95]. Afterwards, maximum likelihood (ML) was inferred using RAxML v8.2.12 [96] (1000 bootstrap pseudo-replicates) under the LG + Γ model. The resulting trees were visualized using the R package ggtree [97], enabling integration with the BGC results.

### 3.7. Antibiotic Resistance Genes 

These analyses were restricted solely to the newly sequenced and assembled genomes of Antarctic bacteria. In detail, antibiotic resistance gene identification was performed using the Resistance Antibiotic Gene Identifier (RGI) tool v6.0.3 (using *--alignment_tool BLAST* and *--include_loose* as parameters) in the Comprehensive Antibiotic Resistance Database (CARD) [98]. Hits showing at least 80% identity with the reference protein were considered significant. Subsequently, each filtered output was processed using a custom Python script to generate two matrices: one indicating the presence or absence of antibiotic-resistant genes, and the other indicating the class of antibiotics to which the specific genome shows resistance. Finally, the obtained data were visualized using the R package pheatmap.

### 3.8. Cold-Shock Proteins

In this study, the presence or absence of a specific set of 15 cold-shock proteins in the sequenced Antarctic bacteria was assessed. The selection of these cold-shock proteins was based on a previous study [71], which utilized an FBH BLAST search on a database comprising 652 bacterial proteins associated with cold adaptation, sourced from the UniProt database.

## 4. Conclusions

Our study underscores the critical importance of sampling diversity in exploring microbial communities in extreme environments such as the Antarctic marine ecosystem. The genetic diversity and functional versatility observed in the newly sequenced genomes highlight the unexploited potential of Antarctic microorganisms as sources of biotechnological products with diverse applications. As a matter of fact, the marine ecosystem presents a promising reservoir of undiscovered bioactive compounds, holding significant potential for biotechnological and pharmaceutical applications [99,100].

Antarctica is thought to be the last pristine continent, characterized by its isolation from external influences due to distinct oceanic and atmospheric circulations [101]. Despite its remoteness and extreme conditions, human activity in Antarctica has steadily risen since the initial documented expeditions in the nineteenth century [102,103]. This translates into an anthropogenic impact, as shown in our study by the presence of genes associated with antibiotic resistance. 

The analyses provided insight into the antibiotic resistance profile of the newly sequenced genomes. Noteworthy was the observation that certain bacteria, such as *Pseudomonas* strains HY13 and 65/3, exhibited a gene repertoire conferring resistance to a wider array of antibiotics (Figure 5).

In our study, we identified genes putatively involved in secondary metabolite production for each of the newly sequenced genomes. Notably, strains belonging to the *Shewanella* and *Pseudomonas* genera displayed a more consistent number of BGCs (Figure 2). These bacterial secondary metabolites represent a valuable resource with diverse applications in biotechnology and pharmaceuticals [104,105]. They play crucial roles as sources of antibiotics, antimicrobial agents, and pharmaceutical products, with additional potential in industrial biotechnology and agricultural biocontrol [104,106,107].

Finally, we delineated the genetic repertoire responsible for cold adaptation. The biotechnological potential of AFPs remains largely undiscovered, with numerous promising applications yet to be exploited [108]. These proteins hold potential in diverse fields, ranging from cryopreservation [109] to CO_2_ hydrate slurry production [110] and even in the preparation of frozen foods [111,112]. From this perspective, *Shewanella* S1-49 is notably well equipped, making it a promising candidate for future biotechnological investigations aimed at fully leveraging its potential.

Overall, our study highlights the genetic and functional diversity of microbial communities in the Antarctic marine environment and provides a foundation for future research aimed at understanding the ecological roles, biotechnological potential, and adaptation strategies of these unique microorganisms. By advancing our knowledge of microbial diversity and ecology in Antarctica, we can better comprehend the resilience of life in extreme environments and contribute to global efforts in conservation and biodiversity preservation.

## Figures and Tables

**Figure 1 marinedrugs-22-00238-f001:**
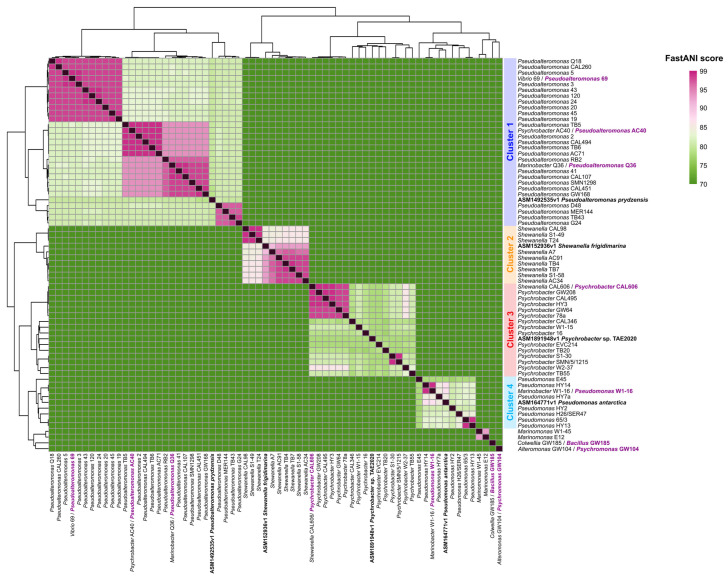
Average nucleotide identity (ANI) represented in a heatmap. Each cell represents a pairwise comparison between the named genomes on rows and columns. Genome names in bold black represent reference genomes obtained from NCBI, while names in bold purple indicate amended taxonomic assignments. Lower values correspond to lower sequence homology and higher phylogenetic distance between strains. The diagonal of the heatmap shows the comparison of each genome with itself, displaying values of 100%. The dendrograms are produced by single-linkage hierarchical clustering trees from the matrix of pairwise identity results.

**Figure 2 marinedrugs-22-00238-f002:**
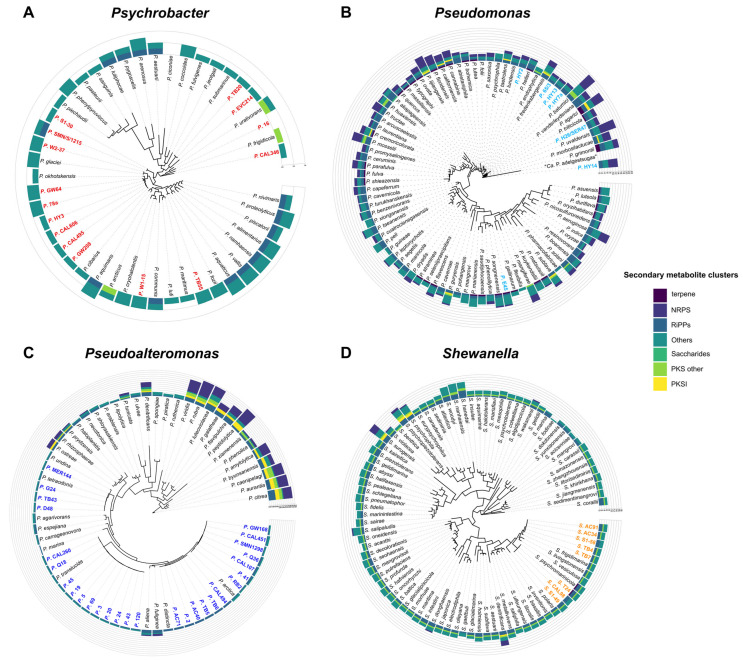
Unrooted circular phylogenomic trees (1000 replicates), reconstructed using the concatenated amino acid sequences of the core genes for each genus. Bold and coloured genome names represent newly sequenced genomes: (**A**) red (*Psychrobacter*), (**B**) light blue (*Pseudomonas*), (**C**) blue (*Pseudoalteromonas*), and (**D**) orange (*Shewanella*). Bootstrap values associated with each tree are specified in Appendix A. The number of biosynthetic gene cluster (BGC) types involved in secondary metabolite biosynthesis are indicated using bar graphs. The BGCs identified were categorized into broader groups using AntiSMASH.

**Figure 3 marinedrugs-22-00238-f003:**
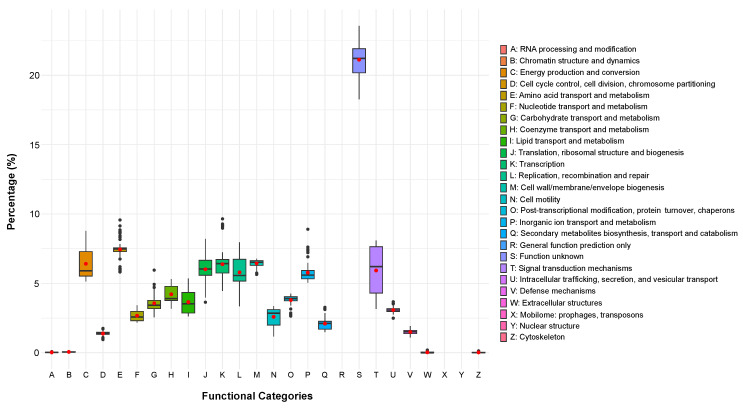
COG functional category distribution across genomes. The red dot within each boxplot shows the mean value of each functional category.

**Figure 4 marinedrugs-22-00238-f004:**
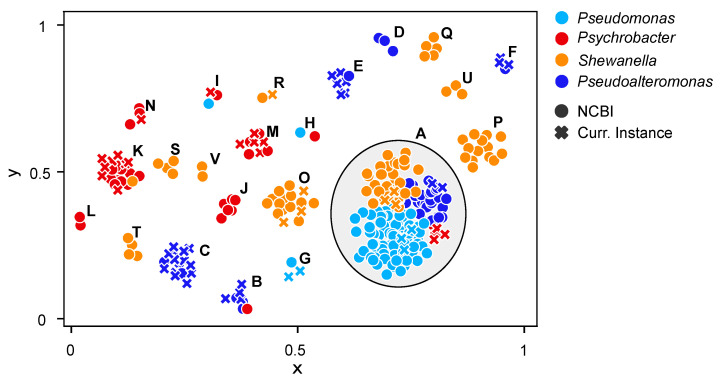
Plot showing the DBSCAN clustering based on the abundance matrix of biosynthetic gene clusters (BGCs) identified in each genome. Each cluster is labeled with a letter of the alphabet. Cluster A, circled in the figure, comprises outliers or data points that did not have enough neighbor points to be part of any cluster. The remaining clusters consist of genomes that exhibit proximity to each other, determined by their similarity. Crosses represent newly sequenced genomes, while circles indicate genomes obtained from NCBI. Four colors are used to distinguish between genera: red (*Psychrobacter*), light blue (*Pseudomonas*), blue (*Pseudoalteromonas*), and orange (*Shewanella*).

**Figure 5 marinedrugs-22-00238-f005:**
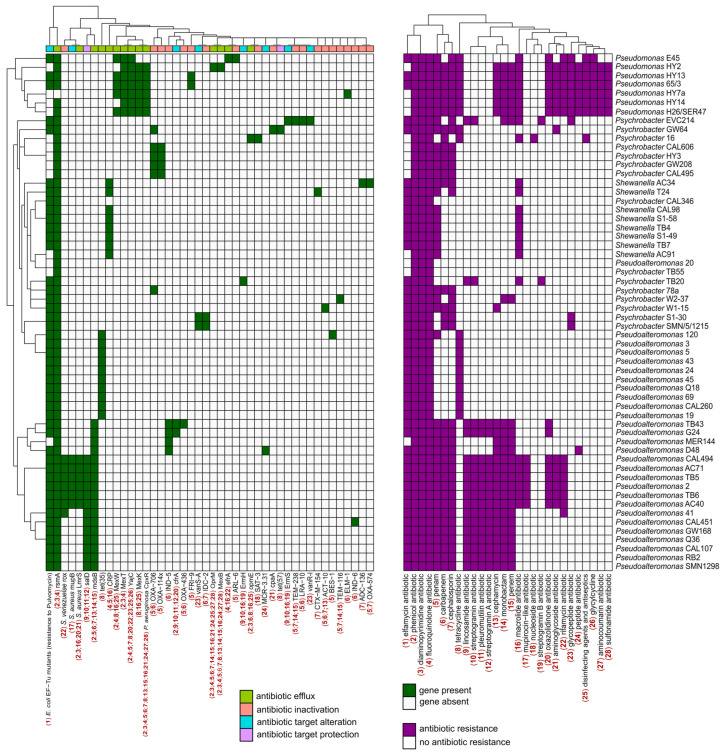
Heatmap representation of the resistome of each genome, as determined by the Resistance Gene Identifier (RGI 6.0.3) tool in the Comprehensive Antibiotic Resistance Database (CARD) version 3.2.9. The left heatmap represents the presence or absence of antibiotic resistance genes in the new genomes, whereas the right heatmap indicates the class of antibiotics to which the genome exhibits resistance. In the heatmap on the right, each antibiotic class is marked with bold red numbers (1 to 28). On the left heatmap, each gene is linked to one or more numbers, indicating resistance to specific classes of antibiotics.

**Figure 6 marinedrugs-22-00238-f006:**
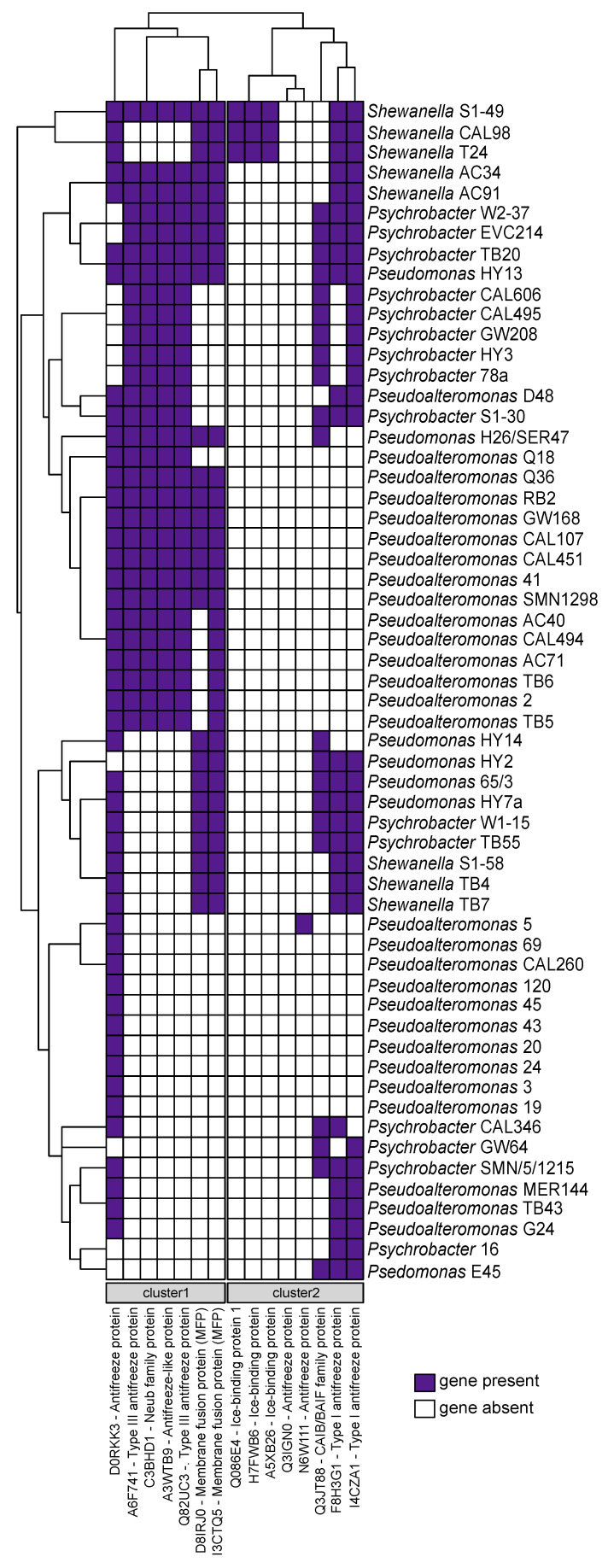
Heatmap of CAPs presence/absence.

**Table 1 marinedrugs-22-00238-t001:** Statistics of the genome assemblies.

	Average Length (Mbp)	Average Number of Contigs	Average % GC	Average Number of CDSs
*Psychrobacter*	4.29	1173	42.8	3394
*Shewanella*	4.96	457	41.7	4171
*Pseudomonas*	6.08	816	59.2	5364
*Pseudoalteromonas*	4.44	694	39.6	3856

## Data Availability

The original genome sequences utilized in this work are available under the NCBI BioProject PRJNA1100444. Sequences falling outside the four designated taxonomic clusters and those considered excessively fragmented have not been deposited on NCBI (see Appendix A).

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
