# Peer review of "Functional Genomics of a Collection of Gammaproteobacteria Isolated from Antarctica"

_marinedrugs, 2024, doi:10.3390/md22060238_

Round 1
Reviewer 1 Report
Comments and Suggestions for Authors
General comments
The paper reports genomic analysis of 62 strains from The Italian Collection of Antarctic Bacteria. This strain collection is of interest by being devoted to microbes from a region of the planet that is difficult to access and, therefore, modestly represented in culture collections. Hence, this whole-genome sequencing effort is a welcomed extension of our knowledge of microbial genomic diversity in this “extremophilic” region.
According to Abstract, the CIBAN is a «repository of cold-adapted bacterial strains isolated from various Antarctic environments». However, the strain selection employed in this study is distinctly more limited, which is not adequately clarified in the manuscript. Firstly, all strains are obtained from a single geographical location (Terra Nova Bay). Secondly, the rather high routine cultivation temperature (21 degrC) indicates that strict psychrophiles, sometimes with Tmax well below 20 degC, are not part of the selection. Thirdly, they are all marine isolates (from seawater, sediment, sponges, one fish species). This is not clarified, neither through title, abstract nor results/discussion; just as a brief mentioning in introduction (possibly the authors find such a clarification unnecessary, given the name and scope of the submitted journal, but I think it is relevant information). All strains appear isolated and routinely cultivated aerobically in marine broth/agar. This approach evidently selects for easily cultivable, relatively fast growing marine bacteria types, which is underscored by the fact that all strains belong to one out of four gammaproteobacterial genera which are all easily isolated from marine environments worldwide. Hence, the paper primarily extends our knowledge of genomic variation within and between established genera of marine Gammaproteobacteria. As such, it consolidates previously observed high levels of heterogeneity in traits like BGCs, resistomes and CAPs among closely related bacteria. On the other hand, the study hardly reveals any distinctive «Antarctic» features in these traits (see comment on title).
Specific comments/suggestions
Title: The title creates the anticipation that the study will disclose some “signatures” that characterize bacterial adaptation to life in Antarctica. The phrase “signatures” is not used except in title, so for this reviewer it is far from obvious what genetic traits that actually constitute these “signatures”. Please clarify this or reformulate title.
Abstract:
L. 32-33: «… while also providing practical solutions for preserving this delicate ecosystem.» I can hardly see in what way the outcome of this study may facilitate better preservation measures in the vulnerable Antarctic environments.
Intoduction:
L. 39-57: I question the necessity of the first two paragraphs of the introductory section. They cover general themes like time of emergence of life on Earth, theoretical estimates of overall global microbial species richness, (unqualified) estimates of fraction of pathogens out of total microbial diversity, and general statements on the use of microbes for various human applications. This background information appears redundant. On the other hand, what could be pointed out – possibly even clearer than in the present manuscript ‑ is the notion that the microbial world still harbours a huge reservoir of untapped functional genomic diversity. In that context, the Antarctica is attractive, as the continent includes extreme environments that expectedly have promoted the evolution of some unique, «extremophilic» adaptations. Its lengthy physical isolation from other continents is likely an additional contributor to exceptional evolutions.
L. 84-90: This characterisation of the Antarctic environment is evidently based on a single reference and appears trivial. It neglects the fact there are significant seasonal, geographical and environmental variations. Since this paper deals with marine bacteria, the marine environments, in particular, deserve a more thorough description of main characteristics and variations.
L. 97: No reference to “lacustrine” samples according to Additional file 1.
L. 102: Suggest … sequencing their 16S rRNA genes and …
Results/Discussion:
L. 162: Suggest …for details on the preparation…»
L. 213: «The number of biosynthetic gene cluster …» The figures have no scales which relate the bars to actual numbers, so just relative numbers are indicated. Please clarify.
Figure 3: The figure appears redundant, as it simply demonstrates the high level of stability in functional category distribution both within and between the different genera. The Antarctic strains are evidently no exception. Hence, these data can just as well be pointed out in text, giving “average” figures and variation ranges.
L. 282-284: The reference [44] does not support these claims.
Conclusions:
L. 496-498: I question that the presence of ARGs «clearly shows» anthropogenic impact, as it is well documented that such genes are found «everywhere», even where human impact is inconceivable.
L. 503: «… we identified secondary metabolites …» This is not precise, as «secondary metabolites» are the potential products of BGC expression. Please rephrase.
L. 505-507: I do not quite follow this argument. Does it imply that the individual strains of Pseudomonas in e.g. Terra Nova Bay seawater has adapted to more «diverse ecological niches» (possibly through HGT) than the Psychrobacter strains in the same environment? If so, please present argument(s). Alternatively, has genus Pseudomonas achieved broader ecological adaptations way back in evolution so that the strains thriving in the Antarctic waters have simply retained this broader, but possibly partly redundant repertoire through subsequent generations? Please clarify.
Comments on the Quality of English LanguageThe manuscript is occasionally marked by phrasings that appear influenced by the authors´ native language.
Reviewer 2 Report
Comments and Suggestions for Authors
The manuscript primarily discusses the genetic diversity and functional versatility of 62 Gammaproteobacteria strains isolated from different Antarctic environments, illuminating adaptation mechanisms through an examination of cold-shock protein distribution. The manuscript presented some novel findings, but there are several concerns that need to be addressed before publication:
1. The names of some resistant genes in Figure 5 are garbled.
2. Line 325, “only hits showing at least 80% identity”, Whether a minimum alignment length was used to screen for positive results.
3. In Fig. 5, the antibiotic resistance profile of these strains is determined based on the presence or absence of the antibiotic resistance genes, right? If so, it is suggested to include the correspondence between antibiotic resistance genes and resistance profiles in the figure.
4. The categories of biosynthetic gene clusters are suggested to be classified into broad categories (PKS Type I, PKS Other types, NRPS, PKS/NRPS Hybrids, Saccharides, Terpenes, RiPPs and Others). The current presentation results make it difficult for readers to see specific information clearly.
5. The distribution of biosynthetic gene clusters in these strains is presented in Figure 2, but the text describing this information is in the next subsection (2.3). It is suggested to reorganize the contents and figures (Figs. 2-4) in Sections 2.2 and 2.3.
6. It is recommended to further analyze the diversity and specificity of the biosynthetic gene clusters encoded by these strains using BIG-SCAPE. This analysis utilizes gene sequence information rather than simple numerical statistical analysis.
7. Genome assembly quality significantly impacts analyses such as Average Nucleotide Identity (ANI), functional genes, and biosynthetic gene clusters. The manuscript should introduce relevant information or discuss potential impacts.
Round 2
Reviewer 2 Report
Comments and Suggestions for Authors
The authors have addressed all my questions and concerns. The manuscript can be accepted in present form.